# Unveiling the structural spectrum of SARS-CoV-2 fusion by in situ cryo-ET

Caner Akıl [1,2,4], Jialu Xu[2,4], Juan Shen[2] & Peijun Zhang [1,2,3] ✉

SARS-CoV-2 entry into host cells is mediated by the spike protein, which drives membrane fusion. While cryo-EM reveals stable prefusion and postfusion conformations of the spike, the transient fusion intermediate states during the fusion process remain poorly understood. Here, we design a near-native viral fusion system that recapitulates SARS-CoV-2 entry and use cryo-electron tomography (cryo-ET) to capture fusion intermediates leading to complete fusion. The spike protein undergoes extensive structural rearrangements, progressing through extended, partially folded, and fully folded intermediates prior to fusion-pore formation, a process that depends on protease cleavage and is inhibited by the WS6 S2 antibody. Upon interaction with ACE2 receptor dimer, spikes cluster at membrane interfaces and following S2' cleavage concurrently transition to postfusion conformations encircling the hemifusion and initial fusion pores in a distinct conical arrangement. S2' cleavage is indispensable for advancing fusion intermediates to the fully folded postfusion state, culminating in membrane integration. Subtomogram averaging reveals that the WS6 S2 antibody binds to the spike's stem-helix, crosslinks and clusters prefusion spikes, as well as inhibits refolding of fusion intermediates. These findings elucidate the entire process of spike-mediated fusion and SARS-CoV-2 entry, highlighting the neutralizing mechanism of S2-targeting antibodies.

The severe acute respiratory syndrome coronavirus 2 (SARS-CoV-2), the causative agent of the COVID-19 pandemic, belongs to the Betacoronavirus genus, which also includes SARS-CoV, MERS-CoV, and several human coronaviruses associated with mild respiratory infections[1-4]. Central to SARS-CoV-2 infection is its spike (S) protein, which facilitates viral entry into host cells[5,6]. The S protein, a homotrimeric class I fusion protein, is cleaved into two subunits, S1 and S2, each with distinct roles in the fusion process[7-9]. The fusion process begins when the S1 subunit binding to the angiotensin-converting enzyme 2 (ACE2) receptor on the host cell surface via its receptor-binding domain (RBD)[10-14]. This interaction triggers a conformational change in the S protein, leading to S1 shedding and exposure of the S2 subunit for fusion activation[10,15]. Initial cleavage of the S protein

occurs at the S1/S2 site via furin protease in the virus-producing cell[16,17]. A second cleavage at the S2' site, mediated by host proteases such as TMPRSS2[18-20] or cathepsin L[21,22] in the target cell, is essential for fusion and infection[16,17,20,22-24].

Class I viral fusion has been extensively studied, with various models proposed[25-29]. Studies of SARS-CoV-2 prefusion and postfusion spike structures reveal significant conformational changes between these states, involving the extension of the S2 hydrophobic fusion peptides (FP), potentially inserting into the host cell membrane, and the refolding of the S protein's C-terminus, including the heptad repeat 2 (HR2) region, into its postfusion conformation[7,8,30-37].

Recently, fusion inhibitory peptides and S2 antibody Fabs were used to lock and stabilize fusion intermediates formed between

[1]Chinese Academy of Medical Sciences Oxford Institute, University of Oxford, Oxford, UK. [2]Division of Structural Biology, Nuffield Department of Medicine, University of Oxford, Oxford, UK. [3]Diamond Light Source, Harwell Science and Innovation Campus, Didcot, UK. [4]These authors contributed equally: Caner Akıl, Jialu Xu. ✉e-mail: peijun.zhang@strubi.ox.ac.uk

pseudo-typed SARS-CoV-2 virus-like particles (VLPs) and ACE2-expressing vesicles[38] or between HIV-1 VLPs and MLV VLPs decorated with spikes or receptors[39]. However, these systems are unable to recapitulate the complete fusion process in native membranes, lacking essential steps and distinct fusion intermediates leading to a full fusion, as well as the effect of S2' cleavage on the fusion process.

Here, we establish a near-native system using authentic infectious SARS-CoV-2 virions and ACE2-expressing HIV-1 VLPs (referred to as ACE2$_{VLP}$s) combined with trypsin treatment for S2' cleavage, which allows a complete membrane fusion and merging of SARS-CoV-2 and ACE2$_{VLP}$s (Fig. 1a). This system enables us to dissect the SARS-CoV-2 spike protein's fusion mechanism by examining its intermediates across nine distinct fusion stages, from receptor binding to complete fusion, as captured by cryo-electron tomography (cryo-ET). Our study directly visualizes the structural spectrum of intermediates in the viral fusion pathway, including S1 prefusion receptor binding mode, extended and partial backfolding S2 intermediates, and the transition to the fully folded S2 conformation during hemifusion and initial fusion pore formation. Cryo-ET and subtomogram averaging further reveal that the S2 stem-helix broadly neutralizing antibody (bnAb) WS6[40] inhibits viral fusion through dual mechanisms: it not only binds to and crosslinks prefusion spikes but also prevents the refolding of S2 by interacting with the extended fusion intermediate. This dual inhibitory action could be a common neutralizing mechanism for this class of S2 stem-helix antibodies.

## Results

### Binding of SARS-CoV-2 spike to ACE2 results in clustering of spike/ACE2 complexes in native membranes

We developed a close-to-native system to study viral fusion using authentic infectious SARS-CoV-2 virions within the biosafety containment, in conjunction with virus-like particles (VLPs) expressing ACE2 on their membrane surfaces (Fig. 1a). Previous studies have shown that spikes on isolated inactivated SARS-CoV-2 virus particles display a range of postfusion conformations, from approximately 3% (chemically fixed) to 75% ($\beta$-propiolactone-inactivated)[41-44]. However, spikes from egressed infectious virus particles in the context of infected cells are exclusively in the prefusion state[45]. To preserve the native spike conformation and avoid artifacts of spontaneous postfusion conversion during virus isolation, we conducted SARS-CoV-2 fusion experiments using egressed infectious virions and ACE2-expressing VLPs directly in tissue culture within a BSL-3 containment lab, prior to fixation and vitrification for cryo-ET analysis (Supplementary Fig. 1a–b). It is noted that while PFA fixation is commonly used to inactivate virus particles and preserve high-resolution protein structures[43], the lipid membranes appear to retain fluidity[46].

We first characterized ACE2 receptor expression in two VLP systems using HIV-1 and murine leukemia virus (MLV) GagPol packaging vectors, respectively (Supplementary Fig. 2). ACE2 proteins were efficiently incorporated into both VLPs, with higher expression in HIV-1 VLPs compared to MLV VLPs (Supplementary Fig. 2a–c). Cryo-EM micrographs of ACE2-expressing VLPs revealed easily identifiable ACE2 molecules, displaying a distinct dimeric "Y" shape, on both HIV-1 (Supplementary Fig. 2d) and MLV (Supplementary Fig. 2e) VLPs, alongside their characteristic capsids: conical for HIV-1 and nearly spherical for MLV. Given the higher ACE2 expression and easily recognizable capsid shape, we used HIV-1 VLPs for subsequent experiments.

Upon addition of ACE2$_{VLP}$s to egressed SARS-CoV-2 virions in tissue culture 24 hours post-infection, interactions between SARS-CoV-2 and ACE2$_{VLP}$ were readily observed, exemplified by one ACE2$_{VLP}$ attracting multiple SARS-CoV-2 particles (Fig. 1b). Closer inspection of the interface between SARS-CoV-2 and ACE2$_{VLP}$ revealed binding between a prefusion spike and an ACE2 dimer (Fig. 1c–d, Supplementary Movies 1–2). The predominant interaction, however, involved

clustered ACE2/spike complexes, accompanied by flattening of the ACE2$_{VLP}$ membrane (Fig. 1e-f, Supplementary Fig. 3a–c). This is consistent with the ability of the dimeric ACE2 receptor to simultaneously bind two spikes, and the trimeric spike's potential to bind multiple ACE2 receptors. Such multiplicity of interactions was also noted in a recent study[39]. The distance between the two membranes in the clustered regions measured on average $16.7 \pm 2$ nm ($n = 18$). No fusion events were observed between SARS-CoV-2 and ACE2$_{VLP}$.

We further assessed the distribution of prefusion and postfusion spikes on SARS-CoV-2 virions in the presence and absence of ACE2$_{VLP}$s (Fig. 1g). Consistent with our previous analysis, the spikes were predominantly in the prefusion conformation in the absence of ACE2$_{VLP}$s, with $24 \pm 12$ prefusion spikes and $0.3 \pm 1$ postfusion spike per virion ($n = 32$).

The addition of ACE2$_{VLP}$s led to a significant reduction in prefusion spikes, with $10 \pm 5$ ($n = 32$) prefusion spikes per virion, and reduced to nearly zero ($0.02 \pm 0.04$, n = 32) when a higher amount of ACE2$_{VLP}$s was used (Fig. 1g).

### Capturing extended and partial backfolding SARS-CoV-2 spike fusion intermediates

Binding of the ACE2 receptor induces an extended conformation of the spike ($n = 20$), where the S1 subunit dissociates, and the S2 fusion peptide inserts into the opposing ACE2$_{VLP}$ membrane (Fig. 2a). This so-called extended intermediate is observed as a cluster of rod-shaped S2 between the two opposing membranes, as reported previously[38,39]. A further intermediate, namely the partial backfolding intermediate, was detected at the periphery of the cluster where the spike cluster starts forming a ring (Fig. 2b). This intermediate features a long density (-19.5–20.5 nm) with a short, kinked branch (-6–7.5 nm), connecting the two membranes (Supplementary Fig. 3d). S2 appears to adopt a partial backfolding conformation. The SARS-CoV-2 and ACE2$_{VLP}$ membranes remain well-separated, with an average distance of $8.3 \pm 3.8$ nm ($n = 4$).

### Complete fusion of SARS-CoV-2 virions with ACE2$_{VLP}$s upon S2' cleavage

To further dissect the fusion mechanism, we added trypsin to the SARS-CoV-2 virion and ACE2$_{VLP}$ mixtures. Trypsin, a serine protease, can replace TMPRSS2 for cleavage at the S2' site, facilitating membrane fusion[38,47]. After trypsin treatment, we observed the S2' cleavage product, which occurred independently of ACE2$_{VLP}$s (Supplementary Fig. 4a).

A band between S2 and S2' was noticed, which appeared consistently across all samples, probably due to antibody cross reactivity. We observed a similar distribution of prefusion ($27 \pm 9.5$) and postfusion ($0.03 \pm 0.018$) spikes in trypsin-treated virions ($n = 32$) compared to the untreated control (Fig. 1g). Cryo-ET confirmed that trypsin treatment alone without ACE2$_{VLP}$ retained the prefusion conformation on the virus surface (Supplementary Fig. 4c). It is only upon receptor binding that the S1 subunits dissociate, exposing the S2' cleavage site (Supplementary Fig. 4a-b).

The two S2 fusion intermediates observed in the absence of trypsin, the extended and partial backfolding intermediates, were also captured in the trypsin-treated samples (Supplementary Fig. 4d–e). Intriguingly, following trypsin addition, several new fusion intermediates were distinctively observed (Fig. 2c–d, Fig. 3), namely the tightly opposing phase, dimpling, hemifusion and, initial fusion pore. During the tightly opposing phase, the SARS-CoV-2 and ACE2$_{VLP}$ membranes became closely aligned, with an average distance of $2.85 \pm 0.25$ nm ($n = 20$) (Fig. 2c–d, Supplementary Fig. 4g). Distinct from the ACE2 binding and S2 extended intermediates, the tightly opposing phase features clustered spike intermediates forming a ring-like structure. The area between the two closely aligned membranes within this ring is notably devoid of apparent density (Fig. 2c–d). At the

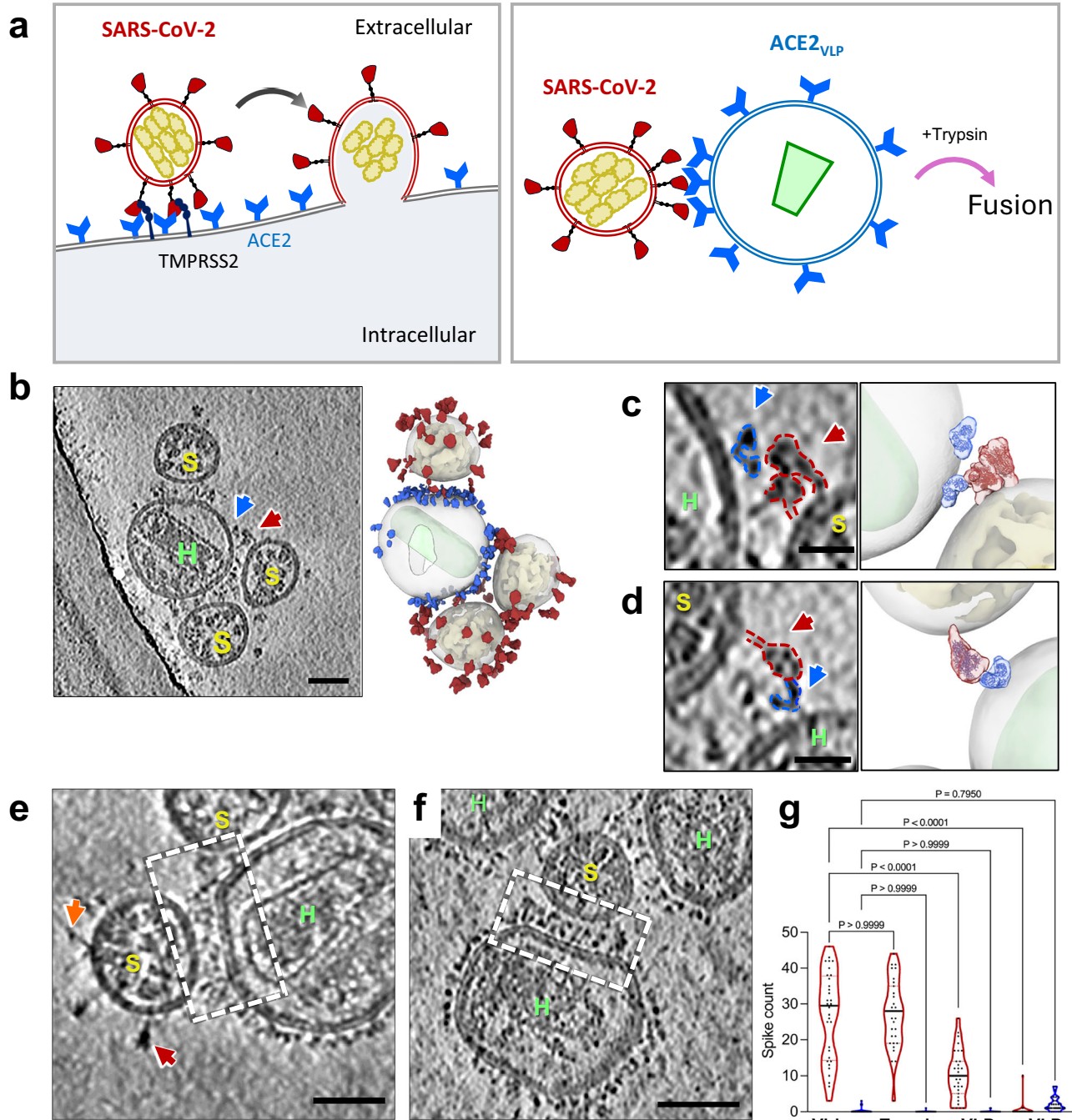

**Fig. 1 | Binding of SARS-CoV-2 spike with ACE2 results in clustering of spike/ACE2 complexes. a** Schematic representation of ACE2-mediated SARS-CoV-2 fusion. Left: Fusion between a SARS-CoV-2 virus and a host cell expressing ACE2 and Transmembrane Protease, Serine 2 (TMPRSS2); Right: Fusion between a SARS-CoV-2 virus and an ACE2$_{VLP}$. Trypsin is added to substitute TMPRSS2. **b** SARS-CoV-2 spike interacting with ACE2$_{VLP}$s, shown in a raw tomographic slice (left), and the corresponding segmented volume (right). SARS-CoV-2 spikes are colored in red, RNPs in yellow, ACE2 dimer in blue, and HIV-1 capsid in light green. **c**–**d** Zoomed-in views of a spike interacting with an ACE2 dimer. **e**–**f** Tomogram slices showing SARS-CoV-2 virions interacting with ACE2$_{VLP}$s, forming predominantly clusters of spike-receptor complexes (dashed white boxes) at low (0.025 mg/ml) (**e**) and high (0.4 mg/ml) (**f**) amount of ACE2$_{VLP}$s. Prefusion spike, red arrow; Postfusion spike, orange arrow. **g** The distribution of prefusion (red) and postfusion (blue) spikes for

SARS-CoV-2 virions alone (left, $n = 32$), with the addition of trypsin ($n = 31$), and with ACE2$_{VLP}$s at low (0.025 mg/ml, $n = 31$) and high (0.4 mg/ml, $n = 31$) concentrations. Data are presented as mean ± SD. Statistical analysis was performed using one-way ANOVA ($F(7, 242) = 119.8$, $p < 0.0001$, $R^2 = 0.7760$), followed by Šídák's multiple comparisons test. Exact $p$ values for pairwise comparisons: for prefusion spike numbers, control vs. virus+trypsin ($p > 0.9999$, ns = not significant), control vs. low concentration ($p < 0.0001$, ****), control vs. high concentration ($p < 0.0001$, ****); for postfusion spike numbers, control vs. virus+trypsin ($p > 0.9999$, ns), control vs. low concentration ($p > 0.9999$, ns), control vs. high concentration ($p = 0.7950$, ns). $n$ indicates the sample size used for statistical analysis. Scale bars: 50 nm (**b**, **e**, **f**); 20 nm (**c**, **d**). ACE2$_{VLP}$s are labeled with "*H*" and SARS-CoV-2 virions with "*S*." Source data are provided as a Source Data file.

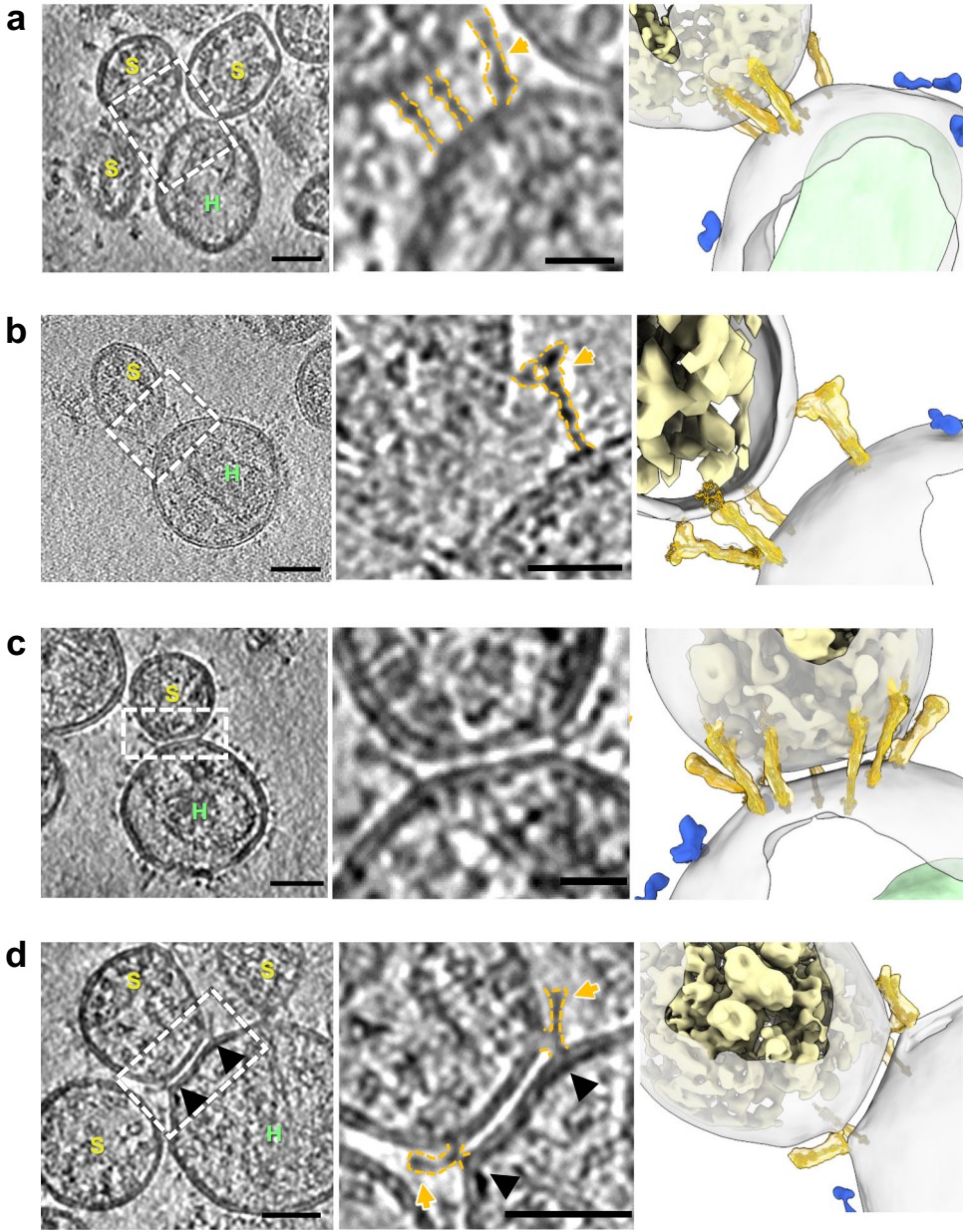

**Fig. 2 | Capturing SARS-CoV-2 fusion intermediates. a** Extended fusion intermediate of SARS-CoV-2 spikes in a tomogram slice (dashed white box), with an enlarged view featuring key structures outlined in gold (middle) and corresponding segmented volumes (right). **b** Partial backfolding intermediate of SARS-CoV-2 spikes in a tomogram slice (dashed white box), outlined in gold in an enlarged view (middle) and their corresponding segmented volumes (right). **c** Trypsin treatment induces the tightly opposing intermediate shown in a tomogram slice (left), an enlarged view (middle) and segmented volume (right), with a protein-depleted bilayer patch which is ring fenced by multiple partial backfolding spikes (right). **d** Spikes in the tightly opposing phase adopt further partial backfolding intermediates, pulling the VLP membrane toward the virion membrane at sharp angles (black arrowheads). Gold arrow indicates spike intermediates. In the segmented volumes, spike intermediates are shown in gold (fitted with a spike model), RNPs in yellow, ACE2 dimers in blue, and HIV-1 capsids in light green. ACE2$_{VLP}$s are labeled "*H*" and SARS-CoV-2 virions "*S*." Scale bars: 50 nm (**a–d**, left) and 20 nm (**a–d**, middle enlarged views).

rim of this membrane patch, spikes adopt deep backfolding conformations (Fig. 2c–d, gold arrows), with the short, kinked branches progressively drawing the VLP membrane towards the virion membrane at a sharp angle (Fig. 2d, black arrowheads. Supplementary Fig. 4f–g). Multiple spikes in a similar deep backfolding intermediate were observed within the same tightly opposing phase, suggesting concerted actions among the spikes (Fig. 2d, Supplementary Movie 3). The tightly opposing phase was the most abundant fusion intermediate observed (183 occurrences from 153 tomograms), and it was only present in the trypsin-treated samples.

Abundant hemifusion intermediates, marked by the merging of the outer leaflets of SARS-CoV-2 and ACE2$_{VLP}$ membranes while their inner leaflets remain separated, were observed exclusively in the presence of trypsin (52 from 153 tomograms) (Fig. 3b, Site 3, Supplementary Fig. 5a, and Supplementary Movie 4). In this state, S2 appeared fully folded, with a density consistent with the postfusion cryo-EM structures (Fig. 3b)[32,41]. The hemifusion membrane necks measured $8.1 \pm 1.1$ nm in diameter ($n = 52$) and were encircled by approximately $5 \pm 1.1$ spikes ($n = 52$), arranged in an apparent postfusion conical configuration (Fig. 3b, Supplementary Movie 4). Furthermore, initial fusion pores, characterized by a further loss of inner leaflet continuity, were also observed, though much less frequently ($n = 3$) (Fig. 3b, Site 4, Supplementary Fig. 4i, 5b, Supplementary Movie 4).

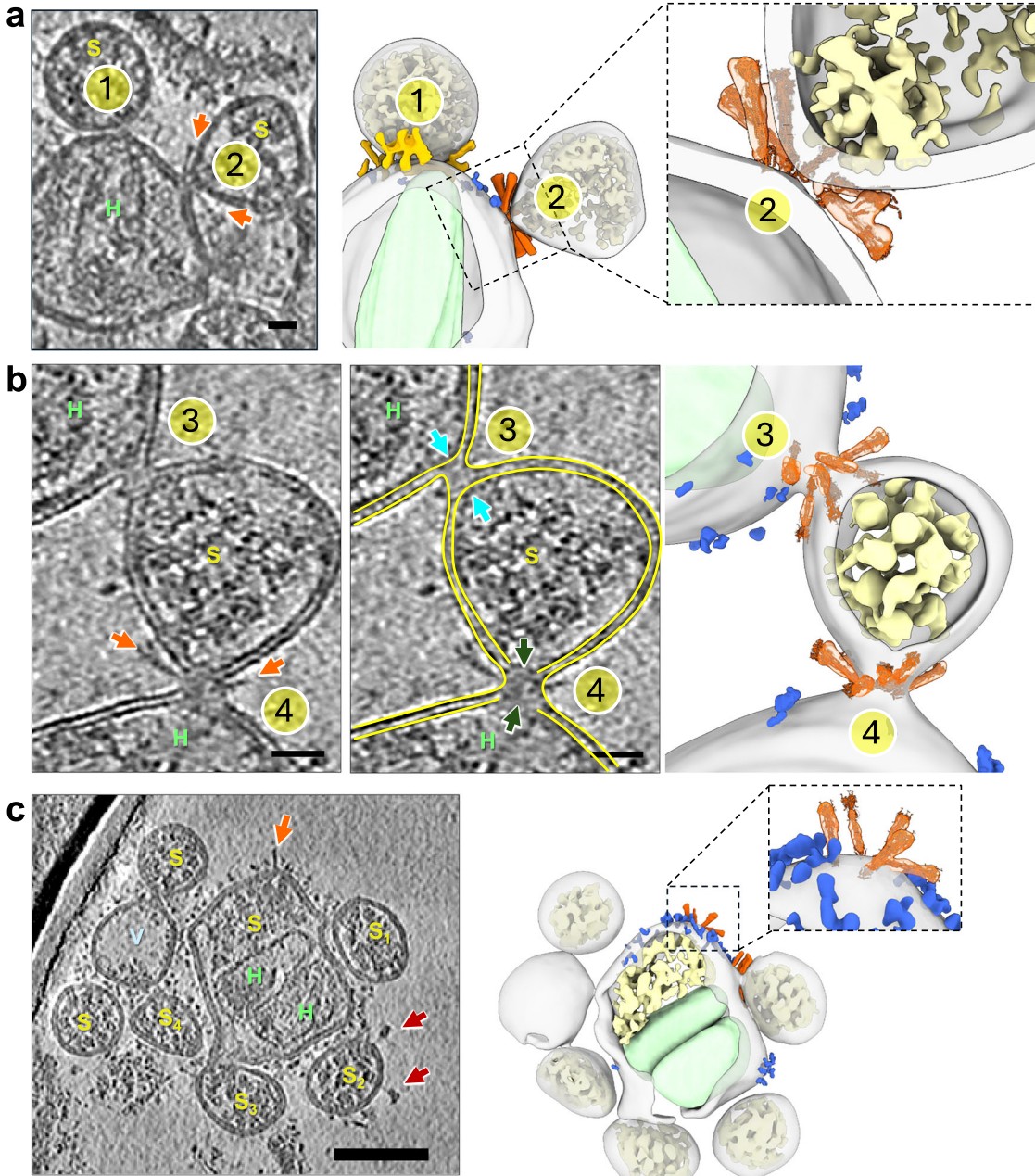

**Fig. 3 | Complete fusion of SARS-CoV-2 virions with ACE2$_{VLP}$s upon trypsin treatment. a** The tightly opposing intermediate (site 1) and the dimpling state (site 2) are shown in a tomogram slice (left) and a segmented volume (middle). An enlarged view of the dimpling state is shown on right (site 2). **b** A hemifusion site (site 3) and an initial fusion pore (site 4) are shown in a raw tomographic slice (left), with corresponding segmentation outlined in yellow (middle), and the segmented volume (right). The differentiation between Sites 3 and 4 as representing distinct fusion states is based on the continuity of the inner leaflet. The inner leaflet appears discontinuous at Site 4 (darkgreen arrows, middle) compared to Site 3 (Cyan arrows). **c** Multiple fusion events between SARS-CoV-2 virions and ACE2$_{VLP}$s in the presence of trypsin, shown with a tomographic slice (left) and the corresponding segmented volume (right). The central particle contains two HIV-1 capsids (light green) associated with SARS-CoV-2 RNPs (yellow), while also engaging in additional fusion processes with four adjacent SARS-CoV-2 particles (S$_1$-S$_4$). ACE2$_{VLP}$s are labeled "H", SARS-CoV-2 virions "S", and vesicles "V." In tomogram slices, SARS-CoV-2 spikes are marked with red arrows (prefusion) and orange arrows (postfusion). In segmented volumes, postfusion spikes are colored in orange and fitted with the postfusion spike model in insets, RNPs in yellow, ACE2 dimer in blue, and HIV-1 capsid in light green. Scale bars: 20 nm in (**a**) and (**b**); 50 nm in (**c**).

Similar to the hemifusion state, multiple postfusion spikes surrounded the initial fusion pore in a conical arrangement. The initial pore membrane necks measured $15.6 \pm 2.2$ nm in diameter ($n = 3$) and were enclosed by an average of $9 \pm 1$ spikes ($n = 3$). While initial fusion pores were rare, numerous fully fused particles ($n = 43$) were identified, containing both SARS-CoV-2 ribonucleoproteins (RNPs) and HIV-1 cores. This suggests that the fusion pore intermediates are transient

and may quickly transition to complete fusion. Evidence of multiple rounds of fusion events was apparent, as large membrane enclosures were observed, decorated with both ACE2 and spikes, and containing multiple HIV-1 cores and numerous SARS-CoV-2 RNPs (Fig. 3c, Supplementary Fig. 6, Supplementary Movie 5). This was further supported by the observation of multiple fusion contacts between SARS-CoV-2 and ACE2$_{VLP}$ (Fig. 3a–c). Fusion scars represented by the residual

ring of postfusion spikes were also seen (Fig. 3c, inset in the right panel). Remarkably, no hemifusion, fusion pore, or fully fused particles were observed in any tomograms ($n = 110$) obtained from samples without trypsin treatment.

### Inhibition of SARS-CoV-2 fusion by S2 antibody WS6

Broadly neutralizing antibodies (bnAbs) are vital for targeting conserved viral epitopes across multiple strains, offering potential cross-protection. One such class of antibodies recognizes the S2 stem-helix region of the spike protein and inhibits viral fusion by blocking the refolding of S2 into the postfusion conformation[39,48–60]. Among these, WS6 has demonstrated remarkable potency and a broad spectrum of neutralization against diverse betacoronaviruses[40].

To investigate the effect of WS6 on SARS-CoV-2 fusion and elucidate its mechanism of inhibition, we added WS6 to tissue cultures containing infectious, egressed SARS-CoV-2 particles. Intriguingly, we observed that prefusion spikes on the virions were linked by a distinct density, likely corresponding to WS6, at the stalk region, forming clusters of crosslinked (or chained) spikes (Fig. 4a, Supplementary Fig. 7a–c). This finding was unexpected, as the previous crystal structure suggested that WS6 binding to the stem helix segment would clash with the prefusion spike conformation[40]. The capacity of WS6 to crosslink spikes could be explained by its two arms binding two separate spikes, akin to the clustering effect observed with ACE2 dimers (Fig. 1e–f). This spike-crosslinking effect likely signifies an additional new mode of action for WS6, further contributing to its potent neutralization of diverse betacoronaviruses.

To confirm the presence of WS6 density at the stalk region of the spike, we determined the in situ structure of the prefusion spike in complex with WS6 at 8 Å resolution using subtomogram averaging (STA) (Supplementary Fig. 8). The atomic model of the WS6 antigen-binding fragment (Fab) (PDB ID: 7TCQ) fits well into the density (Fig. 4b, Supplementary Table 2). Notably, STA and classification revealed that approximately 60% of particles exhibited three WS6 Fabs bound to the membrane-proximal S2 stem (Supplementary Fig. 8).

We also examined the impact of WS6 binding on the flexibility of spikes by comparing WS6-bound spikes with unbound spikes. Statistical analysis of the refined orientations revealed that WS6-bound spikes exhibited reduced tilting around their stalks relative to the viral envelope (Fig. 4c). This suggests that WS6 constrains the hinge flexibility of the spike, thereby modulating its conformational dynamics.

To evaluate the effect of WS6 on SARS-CoV-2 fusion, virions were pre-incubated with WS6 before the addition of ACE2$_{VLP}$s and trypsin. S2' cleavage by trypsin was observed regardless of the presence of WS6 or ACE2$_{VLP}$ (Supplementary Fig. 4b). WS6-bound prefusion spikes as well as WS6-bound extended S2 intermediate were routinely detected (Fig. 4d, purple arrows, Supplementary Fig. 9). Intriguingly, WS6 densities were clearly observed binding to the cluster of extended S2 fusion intermediates approximately midway between the two membranes (Fig. 4d, Supplementary Fig. 9d). However, no partial backfolding intermediates, hemifusion, fusion pore, or fully fused particles were detected in the presence of WS6 across all tomograms ($n = 32$). These findings indicate that WS6 inhibits fusion by binding and clustering both prefusion spikes and extended intermediates, thereby preventing the formation of partial backfolding intermediates.

Notably, WS6 appears not affecting the distribution of pre- and post-fusion spikes on SARS-CoV-2 particles, as indicated by similar spike counts in its absence ($25 \pm 8$ prefusion, $0.5 \pm 0.8$ postfusion) and presence ($24 \pm 12$ prefusion, $0.3 \pm 1$ postfusion) and with additional trypsin treatment ($31.7 \pm 15.7$ prefusion, $0.8 \pm 1.2$ postfusion) without ACE2$_{VLP}$s. However, while the addition of both ACE2$_{VLP}$s and trypsin significantly reduced the number of prefusion spikes ($1.8 \pm 6$), pre-incubation with WS6 provided protection, preserving more prefusion spikes on SARS-CoV-2 particles ($14 \pm 11$) (Fig. 4e).

## Discussion

Capturing the transient and unstable fusion intermediates of class I viral fusion has been challenging. Most studies have focused on influenza A virus (IAV) hemagglutinin (HA)-mediated fusion[61–66]. Under endosomal low pH conditions, HA activation leads to an extended fusion intermediate, as demonstrated using isolated HA[67]. Cryo-ET studies have shown tightly docked membrane contact zones and hemifusion states which ultimately transition to fusion, facilitating viral RNA transfer into liposomes[64,66]. Other class I viral fusion systems, including HIV-1[68–71], murine leukemia virus Env[72], avian sarcoma/leukosis virus (ASLV) Env[73], and SARS-CoV-2[38,39] have been studied. However, most of these studies primarily focused on early fusogen-receptor interactions or stabilizing pre-hairpin-like intermediates using fusion inhibitors. The full process of SARS-CoV-2 class I fusion, from receptor engagement to complete membrane fusion, remains poorly characterized, resulting in incomplete fusion models. We filled in this critical knowledge gap with a full spectrum of fusion intermediates across nine distinct stages through developing and direct imaging a functional class I fusion system.

Based on our results and previous studies, we propose a full model for receptor-mediated SARS-CoV-2 fusion (Fig. 5a–b). In this model, ACE2 binding induces spike clustering (Stage I), which is critical for the subsequent fusion events[68–70,73]. Following receptor binding, S1 dissociates and S2 takes an extended conformation where the FP inserts into the host membrane (Stage II). Subsequently, S2 HR2 begins to refold, forming a partial backfolding S2 intermediate (Stage III). Fusion can proceed only after S2' cleavage, initiating a tightly opposing phase where the two membranes come into close proximity, forming a protein-free membrane patch "ring-fenced" by deep backfolding S2 (Stage IV). S2 refolding progresses to completion, drawing the spikes closer together to form a conical cluster and creating a dimpling phase where the two membranes meet without merging (Stage V). This is followed by the merging of the outer leaflets (hemifusion, Stage VI), the formation of the fusion pore (Stage VII), and complete fusion (Stage VIII). Multiple rounds of fusion produce unusually large particles (Stage IX), Importantly, while WS6 blocks viral fusion at early stages (I and II), fusion halts at Stage III in the absence of S2' cleavage.

Our findings provide three key insights into the fusion process: (1) Receptor-mediated spike clustering and oligomerization plays a critical role in all stages of viral fusion, from initial binding to completion; (2) The ACE2 receptor dimer not only initiates spike clustering but also triggers its conformational changes required for the FP exposure; (3) Cleavage at the S2' site is essential for the transition from the partial backfolding spike intermediates to the fully folded postfusion state. Complete refolding of S2 might be a rate limiting step, as Stage IV is the most abundant intermediate. While we cannot rule out the possibility that the HIV-1 matrix protein on ACE2$_{VLP}$ particles influences the final S2 refolding process (Fig. 3b, Supplementary Fig. 5a), we also observed similar initial fusion pores between a SARS-CoV-2 virion and an ACE2-expressing vesicle that do not contain the HIV-1 matrix protein (Supplementary Fig. 5b).

Spike clustering at membrane interfaces parallels other fusion systems, including SNARE-mediated fusion[74] and HIV-1 Env-CD4 fusion complexes[70], and possibly type II[75–77] and type III[78,79] fusion systems, as well as gamete fusion[80]. The conical assembly of postfusion spikes at the hemifusion and initial fusion pore intermediates resembles the "volcano"-shaped ring of five E1 trimers observed in Semliki Forest virus, suggesting a conserved structural mechanism in viral fusion[81]. Finally, this system allowed us to assess the S2-targeting antibody WS6, revealing its unexpected dual mechanisms of action in potently neutralizing diverse beta-coronaviruses.

While the proposed WS6 mechanism[40] is consistent with other S2-targeting antibodies, such as CC40.8[56], S2P6[57], B6[55], and CV3-25[39], some differences likely arise due to variations in epitope accessibility

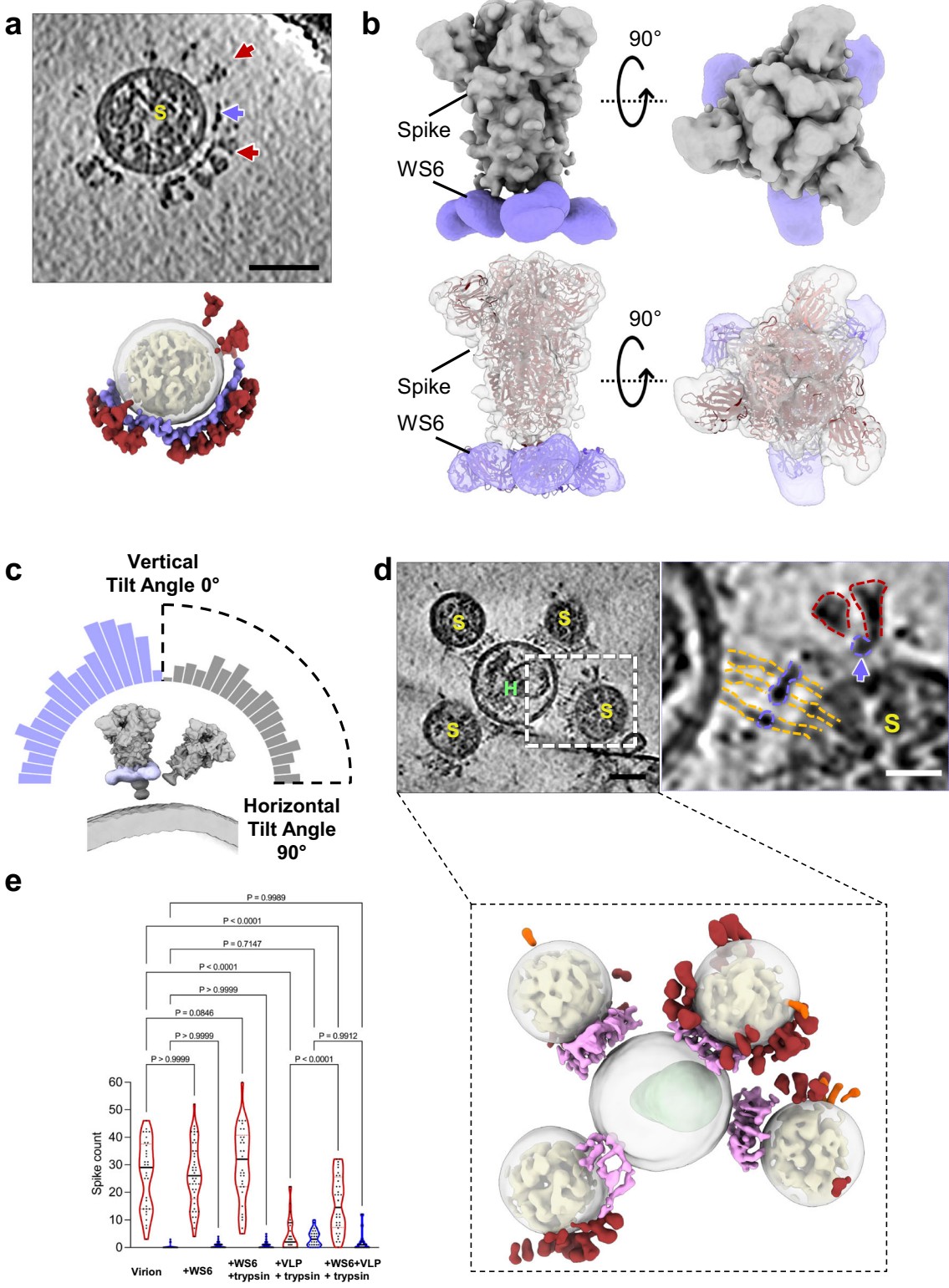

and experimental conditions. Factors such as when the antibody engages the spike protein, whether before or after receptor binding or during fusion, may influence the observed neutralization mechanism. The WS6 antibody, which binds the S2 stem-helix region, induces clustering and crosslinking prefusion spikes and reduces their hinge flexibility (Stage I). WS6 further halts fusion by stabilizing the extended S2 intermediate, preventing S2 refolding (Stage II). These findings

highlight the functional significance of S2 stem-binding antibodies and underscore the critical role of S2 conformational dynamics in successful fusion. Interestingly, we observed that WS6 binds to both state I (prefusion) and state II (extended intermediate) but not to any later states. This is distinct from Grunst et al. [39], which reported an S2 antibody CV3-25 binding to states II and III. This discrepancy likely arises from differences in experimental design: while we pre-incubated

**Fig. 4 | Inhibition of SARS-CoV-2 fusion with a spike stalk antibody WS6. a** WS6 binds and clusters SARS-CoV-2 spikes on the native virions. Shown are tomographic slices of SARS-CoV-2 virions treated with WS6 antibody (top) and the corresponding segmented volume (bottom). WS6 antibody density is shown in purple, prefusion spikes in red, and RNPs in yellow. Purple arrow indicates WS6 and red arrows indicate prefusion spikes. **b** Subtomogram average of WS6-bound prefusion spikes at 9.3 Å resolution (top), with the atomic models of the spike trimer (PDB ID: 6XR8) and three WS6 antibodies (PDB ID: 7TCQ) fitted into the density map (bottom). **c** Distribution of prefusion spike tilt angles relative to the membrane normal axis, for spike alone (gray) and in complex with WS6 (purple). **d** WS6 binding prevents fusion of SARS-CoV-2 with ACE2$_{VLP}$s but retains extended fusion intermediates. Shown are tomographic slices (left, dashed white box) and an enlarged view (right) of SARS-CoV-2 virions in the presence of WS6, with prefusion spikes (red outline) and extended spike intermediates (gold outline), along with the corresponding segmented volume (bottom). Density for fusion intermediates and WS6 between membranes is shown in light purple, prefusion spikes in red,

postfusion spikes in orange, and RNPs in yellow. **e** The distribution of prefusion (red) and postfusion (orange) spikes per SARS-CoV-2 virion for SARS-CoV-2 virion alone ($n = 33$), WS6 antibody ($n = 46$), with trypsin treatment ($n = 34$), with addition of ACE2$_{VLP}$s with trypsin treatment ($n = 31$), and WS6 antibody together with ACE2$_{VLP}$s with trypsin treatment ($n = 32$). Data are presented as mean ± SD. Statistical analysis was performed using one-way ANOVA (F(9, 342) = 82.23, $p < 0.0001$, $R^2 = 0.6839$), followed by Šídák's multiple comparisons test. Exact $p$ values for pairwise comparisons: for prefusion spike numbers, virion vs. +WS6 ($p > 0.9999$, ns = not significant), virion vs. +WS6 + Trypsin ($p = 0.0846$, ns), virion vs. +VLP + Trypsin ($p < 0.0001$, ****), virion vs. +WS6 + VLP + Trypsin ($p < 0.0001$, **); for postfusion spike numbers, virion vs. +WS6 ($p > 0.9999$, ns), virion vs. +WS6 + Trypsin ($p > 0.9999$, ns), virion vs. +VLP + Trypsin ($p = 0.7147$, ns), virion vs. +WS6 + VLP + Trypsin ($p = 0.9989$, ns), +VLP + Trypsin vs. +WS6 + VLP + Trypsin ($p = 0.9912$, ns). $n$ indicates the sample size used for statistical analysis. Scale bars: 50 nm (**a**, **d**) and 20 nm (d, right enlarged view). ACE2$_{VLP}$s are labeled with "*H*" and SARS-CoV-2 virions with "*S*." Source data are provided as a Source Data file.

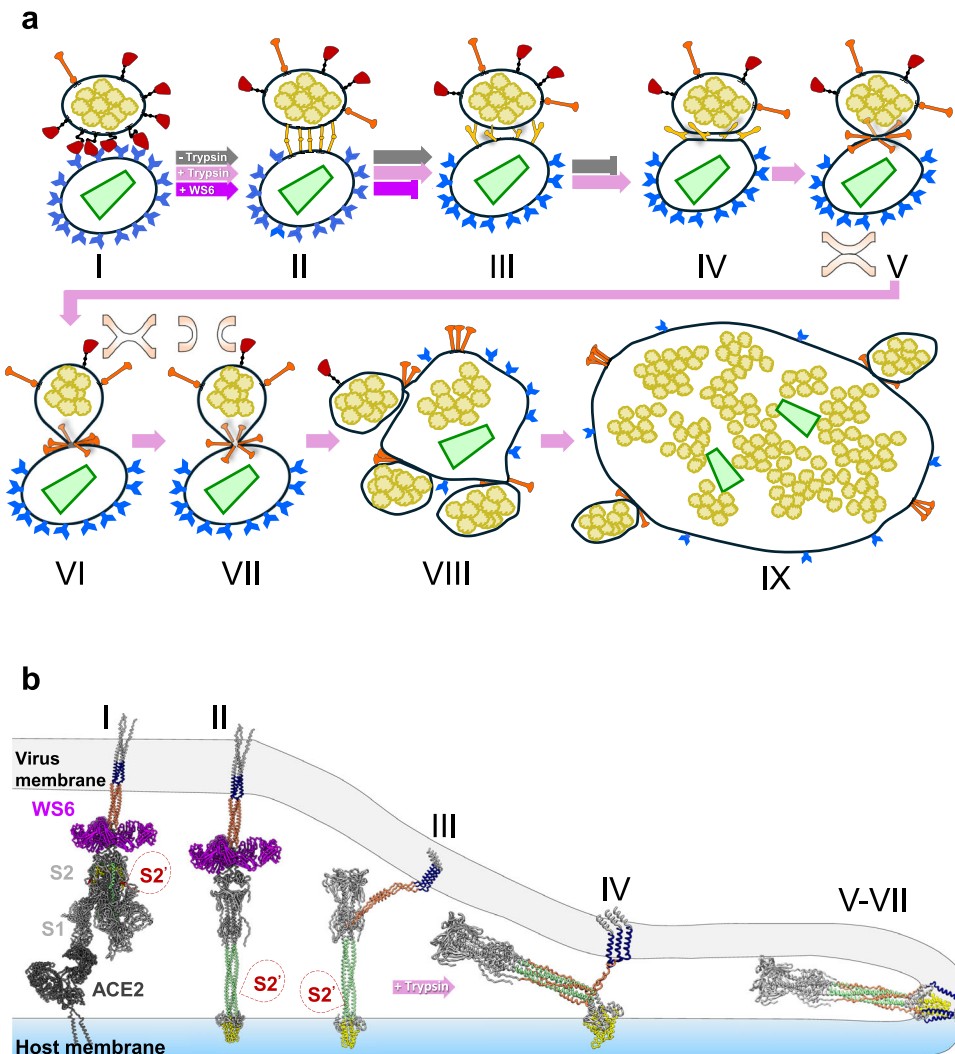

**Fig. 5 | Schematic model of ACE2-mediated SARS-CoV-2 fusion process and fusion intermediates. a** Membrane fusion stages between SARS-CoV-2 and ACE2$_{VLP}$s, including initial binding (I), extended intermediates (II), partial back-folding intermediates (III), tightly opposing intermediates (IV), dimpling phase (V), hemifusion (VI), fusion pore (VII), full fusion (VIII), and multiple rounds of fusion (IX). The HIV-1 capsid is depicted in light green, and RNPs are shown in yellow. The membrane states in V–VII are depicted in inset. Trypsin cleavage of S2' triggers the subsequent processes beyond stage III and completes membrane fusion (IV–VIII) (pink arrows), whereas the WS6 S2 antibody blocks the fusion at stage II (purple

arrows). **b** Models of spike fusion intermediates and the states they are associated with in the schematic above. WS6 antibody binds to spikes in states I and II, inhibiting the refolding of state II intermediate. S2' cleavage triggers a complete refolding from the state III partial backfolding intermediate and the formation of postfusion spike. The image was generated using PDBs 7a97, 6xr8, 8fdw, 6m1d, and 7TCQ. The fusion peptide is shown in yellow, the transmembrane region in navy, heptad region 1 in light green, heptad region 2 in coral, S2' cleavage site marked with a dashed red bubble, and WS6 in purple.

antibodies with virions before adding ACE2$_{VLP}$s, Grunst et al. [39] introduced antibodies after the spike-ACE2 interaction.

Our fusion system provides a comprehensive perspective on spike-ACE2 and spike-antibody interactions, capturing the complete spectrum of fusion intermediates, including many previously unrecognized states. This study not only advances our understanding of SARS-CoV-2 fusion mechanisms and the inhibitory action of S2-targeting antibodies but also holds promise for studying fusion processes across many diverse systems.

## Methods

### Production and Isolation of ACE2$_{VLP}$s

ACE2$_{VLP}$s were produced by co-transfecting human embryonic kidney (HEK) 293 T Lenti-X cells (catalog no. 632180, Takara/Clontech) with HIV-1 GagPol (psPAX2) or MLV GagPol plasmid and a human ACE2-expressing vector (pcDNA) at a 1:1 ratio, using GenJet™ transfection reagent. Two days post-transfection, the culture medium containing the VLPs was collected and filtered through a 0.45 μm filter. The filtered medium was subjected to ultracentrifugation using an SW 32 rotor with a 3 ml cushion of 8% OptiPrep in STE (saline tris-EDTA), centrifuged at 143,726 × g for 1.5 h at 4 °C. The VLP pellet was further purified using an OptiPrep gradient consisting of sequential 1 ml layers of 10, 20, and 30% OptiPrep in STE. Ultracentrifugation was carried out at 245,418.9 × g (SW 55 rotor) for 2.5 h at 4 °C. The ACE2$_{VLP}$ band was carefully collected by puncturing the ultracentrifuge tube at the band's position (Supplementary Fig. 2a). The collected VLPs were subjected to a final centrifugation at 213,787.1 × g (SW 55 rotor) for 1 h at 4 °C. The supernatant was discarded, and the residual buffer was removed by swiping the tube's inner surface with lint-free paper. The final pellet was gently resuspended in 80 μl of STE buffer and stored at 4 °C for use within one day. After purifying the VLPs, the total protein concentration of the purified VLPs was 0.4 mg/ml, measured using the Bradford assay. This stock concentration, normalized for different VLP preparations and referred to as "high concentration," was diluted 16-fold to achieve a "low concentration" of ~0.025 mg/ml.

### SARS-CoV-2 victoria variant culture and titration

SARS-CoV-2 Victoria variant was cultured and handled in a Containment Level 3 (CL3) laboratory at the Oxford Particle Imaging Center (OPIC). The Victoria variant was kindly provided by Tao Dong's group at the CAMS-Oxford Institute. To propagate the virus, Vero E6 cells (6 × 10⁶) were seeded into a T75 flask one day prior. On the day of infection, the culture medium was replaced with 15 mL of DMEM supplemented with 1% FBS and glutamine. Subsequently, 100 μL of the virus, with a titer of 10⁵, was added to the flask and incubated for 2–3 days. After incubation, the culture was centrifuged at 400 × g for 20 min at 4 °C to remove cell debris. The resulting supernatant was aliquoted and stored at −80 °C. The virus titer was determined by plaque assay.

### Western blotting

To confirm ACE2 expression, ACE2$_{VLP}$s were analyzed by western blot using a monoclonal anti-ACE2 antibody (66699-1-Ig, Proteintech; 1:5000 dilution) overnight at 4 °C, followed by an anti-mouse HRP-conjugated secondary antibody (A0168, Sigma-Aldrich; 1:5000 dilution) for 1 h at room temperature (Supplementary Fig. 2b–c).

To assess the effect of trypsin, 30 ml of freshly propagated SARS-CoV-2 Victoria variant was concentrated, and washed twice with PBS using Amicon Ultra centrifugal filters (100 kDa MWCO), resulting in a final volume of 600 μl. 50 μl of concentrated virus was incubated with ACE2$_{VLP}$s at various concentrations, or without ACE2$_{VLP}$s, and with 20 ng/ml trypsin (sequencing grade, V511A, Promega) or without, at 37 °C for 30 min.

For WS6-binding experiments, 50 μl of concentrated virus was incubated with 10 μl of 200 μg/ml WS6 or PBS as a control at 37 °C for

1 h. The virions treated with WS6 or PBS were subsequently incubated with 20 ng/ml sequencing-grade trypsin (V511A, Promega), ACE2$_{VLP}$s, or their combination for 30 min at 37 °C.

SDS-PAGE gel loading dye (2% SDS final concentration) was added to each sample, and the samples were incubated at room temperature for 30 min to inactivate the virus. Samples were then boiled at 99 °C for 10 min and loaded onto an SDS-PAGE gel for western blot analysis using a monoclonal anti-S2 antibody (Thermo Fisher Scientific MA-5-35946) at a 1:4000 dilution. Following primary antibody incubation, the membrane was washed and incubated with a secondary anti-mouse HRP antibody (A0168, Sigma-Aldrich) at a 1:5000 dilution (Supplementary Fig. 4a). To avoid cross-reactivity between WS6, a mouse-derived antibody, and the anti-mouse HRP secondary antibody, western blot analysis for WS6 was performed using a monoclonal anti-S2 antibody (Thermo Fisher Scientific MA5-42384; 1:5000 dilution) with an anti-rabbit HRP-conjugated secondary antibody (CSB-PA564648, CUSABIO). Detection was carried out using Clarity Western ECL substrate solutions (Bio-Rad) (Supplementary Fig. 4b).

### Cryo-ET sample preparation

EM grids (G300F1, R2/2 Quantifoil holey carbon, gold) were glow-discharged and placed in a 12-well plate. The grids were treated with bovine fibronectin (20 μg/ml in PBS) for 30 min, then washed with PBS. Vero E6 cells (7.5 × 10⁴ for Victoria variant) were resuspended in 1 ml of complete DMEM, seeded onto the grids, and incubated at 37 °C with 5% CO$_2$ for 24 h. The plate was then transferred to the CL3 containment facility, and the medium was replaced with SARS-CoV-2 diluted to an MOI of 0.5 in DMEM containing 1% FBS and glutamine. The cells were incubated at 37 °C with 5% CO$_2$ for an additional 24 h. Following incubation, the grids were washed with PBS, and 80 μl of VLPs diluted in PBS were added. For conditions requiring trypsin treatment, 20 ng/mL trypsin (Sequencing Grade, V511A, Promega) in PBS was added simultaneously with VLPs. The grids were incubated for 30 min at 37 °C with 5% CO$_2$, washed with warm PBS, and then fixed with 4% PFA (EM grade) in PBS for 1 h at room temperature before removal from the CL3 facility.

For WS6 antibody-treated conditions, WS6 antibody was diluted to 200 μg/mL in PBS. A total of 500 μL of the antibody solution was added to each well and incubated at 37 °C with 5% CO$_2$ for 1 h. PBS was used as control for the conditions with no WS6. After this incubation, trypsin, ACE2$_{VLP}$s, or their combination were added and further incubated for 30 min at 37 °C. The solution was then removed, and the grids were fixed with 4% paraformaldehyde (PFA, EM grade) in PBS for 1 h at room temperature. The grids were then removed from the CL3 facility for further processing.

### Cryo-ET sample vitrification

PFA-fixed grids were washed with PBS, and 2 μl of PBS containing 6 nm Au fiducial beads (EMS) was applied to the carbon side before blotting. The grids were blotted from the back for 3 s and plunge-frozen in liquid ethane at −183 °C using a Leica GP2 plunger. Subsequently, the grids were clipped and stored in liquid nitrogen until imaging.

### Cryo-ET Data Collection

Tilt series of SARS-CoV-2 were collected using a Titan Krios equipped with a Selectris X energy filter and Falcon 4 detector, or a Gatan BioQuantum energy filter and K3 detector in CDS mode. Tilt angles ranged from −60 to 60 degrees in 3° increments, using a dose-symmetric scheme[82]. Pixel size was 1.50 Å/pixel (Falcon 4 camera) or 1.34 Å/pixel (K3 camera). Each image consisted of 10 movie frames. Cryo-ET data were collected from the periphery of cells containing freshly egressed SARS-CoV-2 particles (see Supplementary Table 1 and 2).

**Cryo-ET data analysis: segmentation and subtomogram averaging.**
Raw movie frames were motion-corrected using MotionCor2[83]. Tilt-series alignment was performed with eTomo[84] based on gold bead fiducial markers or with AreTomo[85] when markers were insufficient. IsoNet neural networks[86] were trained and then used to correct missing wedge artifacts. Virus morphology, including spikes, membrane, ACE2, RNPs and HIV-1 cones, was segmented using MemBrain[87], EMAN2[88], and ChimeraX[89]. Spikes were primarily segmented manually by carefully tracing their shapes on individual tomogram slices, which were subsequently visualized in 3D using ChimeraX to validate. As illustrated in Fig. 1c–d (left) and Fig. 2 (middle), the manually drawn outlines on individual slices highlight the spikes' structure, while the 3D rendering (right panels) emphasizing the identified features.

For Fig. 3 (right), to highlight the details of the membrane bilayer, each leaflet was manually segmented to clearly illustrate their structural transitions. For other components, including the remining membranes, ACE2, RNPs, and HIV cones, segmentation was primarily performed using neural network-based software, including MemBrain and EMAN2, followed by manual cleaning. Automated segmentation was employed for these components to streamline the process.

For subtomogram averaging, CTF estimation was conducted with emClarity[90] modified version of ctffind4[91]. Spikes were identified via emClarity template search and aligned with Relion4[92] for subtomogram averaging. Details of the averaging process are presented in Supplementary figures and Tables. Following subtomogram averaging, spikes were projected onto the membrane based on their refined orientations. The angle between each spike and a perpendicular vector to the membrane, generated from membrane segmentation data, was calculated to represent the tilting angle.

## Quantification and statistical analysis

The representative data are presented as the mean ± SD as indicated and were corrected for multiple comparisons. Statistical analysis was performed using one-way ANOVA followed by post-hoc comparisonswere used to analyze differences in mean values. GraphPad Prism 10.3.0 software was used to calculate the $P$ values, and significance is depicted with asterisks as follows: $*P \leq 0.05$, $**P \leq 0.005$, $***P \leq 0.0005$, $****P \leq 0.0001$. The investigators were not blinded during the experiments or outcome assessment.

## Statistics and reproducibility

The numbers of experiments (tomograms) are listed in Supplementary Table 1 for Figs. 1b–d, 2a–d and 3a–c, and Supplementary Table 2 for Fig. 4a, d. The unmber of observations are indicated in the manuscript.

## Reporting summary

Further information on research design is available in the Nature Portfolio Reporting Summary linked to this article.

## Data availability

All data needed to evaluate the conclusions of this paper are included in the main text and/or Supplementary Information. Source data are provided with this paper. Structural data generated in this study have been deposited in the Electron Microscopy Data Bank (EMDB) under the accession codes: EMD-51333 and EMD-51334. Source data are provided with this paper.

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

## Acknowledgements

We thank Prof. T.B. and F.R. for valuable discussions. The WS6 antibody was generously provided by Dr. T.Z. at the Vaccine Research Center, NIAID/NIH. Vero E6 cells were kindly supplied by Prof. E.F. at the Sir William Dunn School of Pathology, Oxford. We acknowledge Diamond Light Source for access and support of the cryo-EM facilities at the UK National Electron Bio-Imaging Center (eBIC) (proposal NT29812). Computation was performed at the Oxford Biomedical Research Computing (BMRC) facility, supported by the Wellcome Trust Core Award Grant Number 203141/Z/16/Z with additional support from the NIHR Oxford BRC. This work was supported by the Chinese Academy of Medical Sciences (CAMS) Innovation Fund for Medical Science (CIFMS), China (grant no. 2018-I2M-2-002, PZ), the National Institutes of Health grant P50AI150481 (PZ) and R21AI184080 (PZ), the UK Wellcome Trust Investigator Award 206422/Z/17/Z (PZ), the UK Wellcome Discovery Award 311427/Z/24/Z (PZ), and the ERC AdG grant 101021133 (PZ).

## Author contributions

C.A. and P.Z. conceived the study. C.A. and P.Z. designed the experiments. J.S. and C.A. prepared the cryo-EM samples. C.A. and J.S. performed functional validation analysis. C.A. and J.X. collected the data. C.A., J.X., and P.Z. analyzed the data. C.A., J.X., and P.Z. wrote the manuscript with contributions from all authors.

## Competing interests

The authors declare no competing interests.
