## [Transparent Peer Review file · Nature Communications]

Unveiling the Structural Spectrum of SARS-CoV-2 Fusion by In Situ Cryo-ET

Corresponding Author: Professor Peijun Zhang

Version 0:

Reviewer comments:

Reviewer #1

(Remarks to the Author)

The authors have well responded to my previous concerns. There are not many high resolution fusion intermediate structures of viral glycoproteins. Caillat et al eLife 2021 reported a HIV-1 gp41 structure that represents stage 5 reported in this paper. It is up to the authors discretion to add this in their discussion to support a general mechanism for class I fusion proteins.

Reviewer #2

(Remarks to the Author)

I reviewed an earlier version of this study that was submitted to another sister journal and have been asked to respond to the author response to reviewer comments. I appreciate the authors' efforts to clarify the points that were raised, but some sticking points remain in my view. My primary concerns relating to the previous manuscript related to the interpretation and assignment of membrane fusion intermediate states and the possible effects resulting from the rather prolonged chemical fixation step that was used to inactivate the infectious sample (1 hour incubation with 4% PFA at room temperature). On this last point, the authors note that protein structure, specifically for SARS-CoV-2 spike has been shown to be preserved under similar fixation conditions. On this I fully agree. The area of my concern, however, is that the dynamic membrane fusion intermediate states involving refolding proteins and fluid lipid membranes are much more likely to reorganized during the prolonged fixation step, resulting in possible membrane, virus and protein features that may not reflect the on-pathway intermediate states.

Areas that relate to these concerns are in Figure 3 and related supplemental figures and movies discussed on lines 173-195. For example: "Abundant hemifusion intermediates, marked by the merging of the outer leaflets of SARS-CoV-2 and ACE2VLP membranes while their inner leaflets remain separated, were observed exclusively in the presence of trypsin (52 from 153 tomograms) (Figure 3b, Site1, Supplementary Movies 4)." I believe the authors meant Figure 3b, site 3, which is presented as an example of hemifusion. I'm afraid I can't see how sites 3 and 4 in Fig3b are qualitatively different from each other, though in the legend site 3 is designated "hemifusion" and site 4 is called an "initial pore". As shown in the figure, both sites appear to have contiguous inner leaflet densities. The additional examples provided in Supplementary Figure 4h,i and 5, which the authors attribute to hemifusion and initial pore types of contacts are similarly difficult to clearly interpret as the density for the leaflets and other material at the contact site in the supplemental figure example are not straightforward to trace.

Attributing these types of contact sites to on-pathway fusion intermediate states seems difficult due to the prolonged, room temperature chemical fixation step as noted above. For example, could internal viral proteins be crosslinked together, leading to restriction on membrane reorganization or relaxation? Are there crosslinks between lipids and fusion proteins or other proteins that are present that could alter the structures at these sites? How does the membrane relax or reorganize during the 1 hour incubation even as crosslinking is taking place?

Accurately identifying the type of contact including interpreting membrane organization seems to be of key importance to

using cryo-electron tomography to study how membrane fusion takes place. Of the 53 other examples that were assigned as hemifusion, perhaps there are clearer examples?

Due to these caveats, in my view, the title “Unveiling the Complete Spectrum of SARS-CoV-2 Fusion Stages by In Situ Cryo-ET” and statements such as “These findings elucidate the complete process of spike-mediated fusion and SARS-CoV-2 entry...” or “full spectrum of intermediates in the viral fusion pathway...” make a bigger claim than the data definitively support, specifically given the ambiguity introduced by prolonged chemical fixation step and the somewhat less than definitive interpretation of membrane organizations in the examples provided.

Minor item: In the section referencing clustering of other fusion proteins, lines 292-294, suggest to add references for type II (PMID 23898184 and 35970990) as well as type III (PMID 39630917) fusion systems.

Point-by-point responses to reviewers' comments

Reviewer #1 (Remarks to the Author):

The authors have well responded to my previous concerns. There are not many high resolution fusion intermediate structures of viral glycoproteins. Caillat et al eLife 2021 reported a HIV-1 gp41 structure that represents stage 5 reported in this paper. It is up to the authors' discretion to add this in their discussion to support a general mechanism for class I fusion proteins.

-Thank you for the suggestion; we have now included this (page 6).

Reviewer #2 (Remarks to the Author):

I reviewed an earlier version of this study that was submitted to another sister journal and have been asked to respond to the author response to reviewer comments. I appreciate the authors' efforts to clarify the points that were raised, but some sticking points remain in my view. My primary concerns relating to the previous manuscript related to the interpretation and assignment of membrane fusion intermediate states and the possible effects resulting from the rather prolonged chemical fixation step that was used to inactivate the infectious sample (1 hour incubation with 4% PFA at room temperature). On this last point, the authors note that protein structure, specifically for SARS-CoV-2 spike has been shown to be preserved under similar fixation conditions. On this I fully agree. The area of my concern, however, is that the dynamic membrane fusion intermediate states involving refolding proteins and fluid lipid membranes are much more likely to be reorganized during the prolonged fixation step, resulting in possible membrane, virus and protein features that may not reflect the on-pathway intermediate states.

We thank the reviewer for raising the point regarding the potential influence of PFA fixation on the membrane fusion and would like to respond as below:

1. First, our study is based on comparative analyses: All samples, regardless of trypsin treatment or WS6 antibody presence, underwent the same fixation protocol and were processed identically. Therefore, any observed differences among treatment groups are unlikely to result from PFA fixation.
2. A substantial body of literature indicates that PFA primarily cross-links proteins while exerting limited influence on the overall fluidity of lipid bilayers. For instance, Tanaka et al. (Nature Methods, 2010) demonstrated that glycosylphosphatidylinositol (GPI)-anchored proteins remain mobile in the plasma membrane after treatment with 4% PFA at 25 °C for 30 minutes. This suggests that the lipid bilayer retains significant fluidity under these conditions. Extending the fixation time to 1 hour, as in our study, is unlikely to significantly alter this behaviour.
3. Given the evidence that lipid membranes likely remain fluid during fixation, the effect of 1-hour PFA treatment may be comparable to an extended incubation period, additional to the 30-minute prior to the PFA fixation. While fluid lipid membranes were allowed to reorganize during 1.5-hr time frame before cryo-fixation, our analysis captures an ensemble of fusion intermediates without assigning them to specific time points.

Nonetheless, we acknowledge that chemical fixation imposes certain limitations. In response to the reviewer's concern, we have revised the manuscript to explicitly address this limitation (page 3).

Areas that relate to these concerns are in Figure 3 and related supplemental figures and movies discussed on lines 173-195. For example: "Abundant hemifusion intermediates, marked by the merging of the outer leaflets of SARS-CoV-2 and ACE2VLP membranes while their inner leaflets remain separated, were observed exclusively in the presence of trypsin (52 from 153 tomograms) (Figure 3b, Site1, Supplementary Movies 4)." I believe the authors meant Figure 3b, site 3, which is presented as an example of hemifusion. I'm afraid I can't see how sites 3 and 4 in Fig3b are qualitatively different from each other, though in the legend site 3 is designated "hemifusion" and site 4 is called an "initial pore". As shown in the figure, both sites appear to have contiguous inner leaflet densities. The additional examples provided in Supplementary Figure 4h,i and 5, which the authors attribute to hemifusion and initial pore types of contacts are similarly difficult to clearly interpret as the density for the leaflets and other material at the contact site in the supplemental figure example are not straightforward to trace.

Thank you for catching the error. We have corrected it.

We appreciate the reviewer's insightful question. We agree that, for general readers, it may not be straightforward to distinguish these sites in raw tomograms. The differentiation between Site 3 and Site 4 as representing distinct fusion states in Figure 3b is based primarily on the continuity of the inner membrane leaflet.

At Site 4, the inner membrane appears discontinuous compared to Site 3, as highlighted in the segmented membrane density shown in the middle panel of Figure 3b. The background densities, possibly corresponding to the HIV-1 matrix protein, were observed in place of the inner leaflet at Site 4. A similar state displaying discontinuous inner leaflet was captured when a SARS-CoV-2 virion fusing with a ACE2-coated vesicle, which showed reduced background densities (Supplementary Figure 5b, Site 4). These observations led us to interpret Site 4 as an intermediate state between hemifusion and a classical fusion pore. We refer to it as an "initial fusion pore," acknowledging that this assignment is based on a hypothetical model of fusion pore (merging of inner leaflets), as a definitive image of a clear fusion pore has yet to be captured.

We recognize that the difference in membrane continuity may not be readily apparent to all readers. We have revised figure legend to more clearly describe this distinction and included additional supplementary figures (Supplementary Figure 4i and 5a) to further illustrate these states.

Attributing these types of contact sites to on-pathway fusion intermediate states seems difficult due to the prolonged, room temperature chemical fixation step as noted above. For example, could internal viral proteins be crosslinked together, leading to restriction on membrane reorganization or relaxation? Are there crosslinks between lipids and fusion proteins or other proteins that are present that could alter the structures at these sites? How does the membrane relax or reorganize during the 1 hour incubation even as crosslinking is taking place?

We appreciate the reviewer's comment. The internal viral proteins could be crosslinked, but lipid membranes likely remain fluid. Please see our detailed response above regarding the potential influence of PFA fixation on membrane and protein mobility.

Accurately identifying the type of contact including interpreting membrane organization seems to be of key importance to using cryo-electron tomography to study how membrane fusion takes place. Of the 53 other examples that were assigned as hemifusion, perhaps there are clearer examples?

The hemifusion sites across the tomograms exhibit comparable quality. We have included additional Supplementary Figure 4i and 5a to further illustrate these states.

Due to these caveats, in my view, the title "Unveiling the Complete Spectrum of SARS-CoV-2 Fusion Stages by In Situ Cryo-ET" and statements such as "These findings elucidate the complete process of spike-mediated fusion and SARS-CoV-2 entry..." or "full spectrum of intermediates in the viral fusion pathway..." make a bigger claim than the data definitively support, specifically given the ambiguity introduced by prolonged chemical fixation step and the somewhat less than definitive interpretation of membrane organizations in the examples provided.

In light of the reviewer's suggestion, we have revised the title to: 'Unveiling the Structural Spectrum of SARS-CoV-2 Fusion by In Situ Cryo-ET' and other relevant contents in the manuscript.

Minor item: In the section referencing clustering of other fusion proteins, lines 292-294, suggest to add references for type II (PMID 23898184 and 35970990) as well as type III (PMID 39630917) fusion systems.

Thank you for this suggestion; we have now included these references.